# Psychometric properties of a screening tool for autism in the community—The Indian Autism Screening Questionnaire (IASQ)

Satabdi Chakraborty[1], Triptish Bhatia[2], Vikas Sharma[3], Nitin Antony[4], Dhritishree Das[4], Sushree Sahu[5], Satyam Sharma[6], Vandana Shriharsh[6], Jaspreet S. Brar[7], Satish Iyengar[8], Ravinder Singh[9], Vishwajit L. Nimgaonkar[10], Smita Neelkanth Deshpande[11]*

1 Department of Psychiatric Social Work, Centre of Excellence in Mental Health, Atal Bihari Vajpayee Institute of Medical Sciences-Dr. Ram Manohar Lohia Hospital, New Delhi, India, 2 Indo-US Projects and NCU-ICMR, Department of Psychiatry and De-addiction, Centre of Excellence in Mental Health, Atal Bihari Vajpayee Institute of Medical Sciences-Dr. Ram Manohar Lohia Hospital, New Delhi, India, 3 NCU-ICMR, Department of Psychiatry and De-addiction, Centre of Excellence in Mental Health Atal Bihari Vajpayee Institute of Medical Sciences-Dr. Ram Manohar Lohia Hospital, New Delhi, India, 4 Development and Validation of the Screening Version of ISAA, ICMR Project', Dept. of Psychiatric Social Work, Centre of Excellence in Mental Health, Atal Bihari Vajpayee Institute of Medical Sciences-Dr Ram Manohar Lohia Hospital, New Delhi, India, 5 Dyslexia Assessment for Languages of India, MoST Project, Department of Psychiatry and De-addiction, Centre of Excellence in Mental Health, Atal Bihari Vajpayee Institute of Medical Sciences-Dr Ram Manohar Lohia Hospital, New Delhi, India, 6 Department of Clinical Psychology, Centre of Excellence in Mental Health, Atal Bihari Vajpayee Institute of Medical Sciences-Dr. Ram Manohar Lohia Hospital, New Delhi, India, 7 Department of Psychiatry and Consultant, Community Care Behavioral Health Organization, Western Psychiatric Hospital of UPMC, Pittsburgh, Pennsylvania, United States of America, 8 Department of Statistics, University of Pittsburgh, Pittsburgh, Pennsylvania, United States of America, 9 Non Communicable Diseases, Indian Council for Medical Research, New Delhi, India, 10 Department of Psychiatry and Department of Human Genetics, University of Pittsburgh School of Medicine and Graduate School of Public Health, Pittsburgh, Pennsylvania, United States of America, 11 Department of Psychiatry, Centre of Excellence in Mental Health, Atal Bihari Vajpayee Institute of Medical Sciences-Dr. Ram Manohar Lohia Hospital, New Delhi, India

☉ These authors contributed equally to this work.
* smitadeshp@gmail.com

**Data Availability Statement:** All relevant data are within the paper and its Supporting Information files.

## Abstract

### Introduction

Currently available screening questionnaires for Autism spectrum disorders were tested in developed countries, but many require additional training and many are unsuitable for older individuals, thus reducing their utility in lower/ middle- income countries. We aimed to derive a simplified questionnaire that could be used to screen persons in India.

### Methods

We have previously validated Indian Scale for Assessment of Autism (ISAA), that is now mandated for disability assessment by the Government of India. This detailed tool requires intensive training and it is time consuming. It was used to derive a new screening questionnaire: 1) items most frequently scored as positive by participants with autism in original ISAA validation study were modified for binary scoring following expert review. 2) In a new

**Funding:** This work is supported by the Indian Council Medical Research (ICMR) under Capacity Building Projects for National Mental Health Programme (ICMR-NMHP) Task Force and 'Cross-Fertilized Research Training for New Investigators in India and Egypt' (D43 TW009114, HMSC File No. Indo-Foreign/35/M/2012-NCD-1, funded by Fogarty International Centre, NIH). The content of this manuscript is solely the responsibility of the authors and does not necessarily represent the official views of NIH or ICMR. NIH and ICMR had no role in the design and conduct of the study; collection, management, analysis, and interpretation of the data; preparation, review, or approval of the manuscript; and decision to submit the manuscript for publication.

**Competing interests:** The authors have declared that no competing interest exist.

sample, clinically diagnosed individuals with/without autism were administered the screening tool and ISAA following written informed consent. Its psychometric properties were determined.

## Results

A 10-item scale named Indian Autism Screening Questionnaire (IASQ) was prepared in Hindi and English. Thereafter 145 parents/caregivers of participants (autism, n = 90, other psychiatric disorders, n = 55) (ages 3–18), were administered IASQ and ISAA (parents/caregivers plus observation) by separate interviewers, blind to each other and to diagnosis. At a cutoff of 1, sensitivity was 99%, specificity 62%, Positive Predictive Value 81%, and Negative Predictive Value 95%. Test-retest reliability was r = 0.767 (CI = 0.62–0.86) and interrater reliability- Krippendorff"s-alpha was 0.872. The area under Receiver Operating Characteristic Curve (ROC) was 95%. There was a significant difference on IASQ-scores between participants with and without a clinical diagnosis of Autism (t = 14.57, p<0.0001).

## Discussion

The IASQ is a simple, easy to use screening tool with satisfactory reliability and validity, that can be administered to caregivers in 15 minutes and provides information about DSM 5 criteria for autism. It may be applicable outside India, following additional adaptation, for community-based studies.

## Introduction

Autism Spectrum Disorder (ASD) is a neurodevelopmental disorder manifesting at an early age, persisting throughout life. It is a significant public health concern in India with estimated 1.7–2 million affected children [1]. Its prevalence ranges from 0.15% to 1.01% depending on the screening method used [2–5]. Screening for ASD is important in order to expedite early interventions to improve behavior and for education and training [6, 7]. Care-giver report screening could be the best method for early identification of autism in low- and middle-income (LMIC) countries, due to their large populations and inadequate healthcare resources [8].

An autism screening tool is a brief questionnaire or checklist completed by a parent or caregiver for preliminary identification of a child with possible autism [9]. Definitive diagnosis can be made after detailed clinical evaluation by experienced and trained clinicians—the gold standard for diagnosis [8, 10]. Rashtriya Bal Swasthya Karyakram (RBSK) is a flagship Government of India Programme focused on early identification and intervention for children from birth to 18 years. It includes early identification of 4 'D's viz. Defects at birth, Deficiencies, Diseases, Development delays including disability (including autism) [11]. For early identification all children should be screened in the community at a younger age. Community screening has several advantages [12, 13]. Diagnosis of autism is often delayed [14, 15]. After India signed the United Nations Convention on the Rights of Persons with Disabilities—UNCRPD [16], detecting autism at both younger and older ages became important so that all eligible persons could avail of disability concessions and benefits. But the usefulness of screening tools for older children and adults is largely unstudied [14].

## Internationally developed screening tools

Most autism screening tools developed and tested in Western societies focus on infants and young children. Only six screening instruments were developed especially for use in low and middle income countries [17]. Among these, three commonly used tools are the Modified Checklist for Autism in Toddlers (M-CHAT), [18] Revised with follow up, the Pictorial Autism Assessment Schedule (PAAS) [19, 20] and the Three Item Direct Observation Screen (TIDOS) [21]. The age range for M-CHAT is 16–30 months, for PAAS- 18–48 months and for TIDOS- 18–60 months. The sensitivity and specificity for all the three tools is reportedly above 70%. But these screening tools have been developed and studied in infants and young children. For adults and older children, only Autism Spectrum Quotient or its shorter version (AQ-50, AQ-S), Ritvo Asperger and Autism Diagnostic Scale-Revised (RAADS-R and RAADS-14) are said to have satisfactory or intermediate values for their psychometric properties with strong or moderate evidence [22]. Commonly used screening tools in India are the Childhood Autism Rating Scale (CARS), and the Modified Checklist for Autism in Toddlers (M-CHAT) [18, 23] which cannot evaluate older persons.

A screening tool should be designed so that it can be used by primary health care workers with minimum training to identify possible at-risk individuals in the general population. There is also a need for validation of these tools in older children and adults [22]. The utility of a screening tool depends on its sensitivity, specificity and generalizability in the population in which it is intended to be used [24–27]. In LMIC an ideal tool—with good psychometric properties administered by minimally trained workers—should screen and identify both younger and older persons [28].

## Indian screening tools

The INCLEN Diagnostic Tool for Autism Spectrum Disorder (INDT-ASD) was developed for 2-9-year-olds but cannot be used in the community by non-specialists [29]. Some authors inappropriately used the Hindi version of the Indian Scale for Assessment of Autism (ISAA) as a screening tool by clinical psychologists to report a community prevalence of 0.9/1000 [30]. For children between 1.5–6 years old, a screening tool, developed through a survey, demonstrated a high diagnostic hit rate of 0.89, specificity of 0.90 and sensitivity of 0.88 [31]. The Chandigarh Autism Screening Instrument (CASI) is a 37-item questionnaire-based tool designed for community screening for autism for children aged 1.5–10 years [32]. A shorter version is also available.

The Indian Scale for Assessment of Autism (ISAA) was developed by the Government of India for detailed assessment of autism and is mandated for evaluating degree of disability. It is a 40-item tool which can be used for certification and follow-up but **not** for screening [33]. ISAA was developed and validated in a multi-centric Pan-Indian study comprising 1124 participants aged 3–22 years with clinically diagnosed Autism/Pervasive Developmental Disorder (PDD-ICD10), Intellectual Disability (ID), other psychiatric disorders and persons with normal intellect. The scale was compared with Childhood Autism Rating Scale (CARS) (Version 1) [12]. Reliability and internal consistency of ISAA as compared to CARS was satisfactory [12, 33]. Validity of ISAA was determined through content validity and criterion validity. Initially an item pool of 437 items was generated, shortened to 57 after expert inputs from 30 experts, and then to 40 after a pilot study. At the end of the main study, each individual item was correlated with total ISAA scores which was significant at 0.001 level suggesting that all items were valid in differentiating between autism and other groups. The autism group had significantly higher scores than other groups suggesting high discriminant value. The criterion test validity of ISAA was determined by comparing total scores on ISAA with those on CARS using

Pearson Product Moment Correlation (r = 0.765, p<0.001). Internal consistency and reliability (Cronbach's coefficient alpha) were significant and comparable to CARS (Cronbach's alpha 0.932 P < 0.001) [12]. However, the ISAA is a detailed evaluation tool which takes time and training, and cannot be used for population studies.

As part of a Capacity Building Task Force project of the Indian Council of Medical Research, a study was planned to develop a short, easy-to-use community screening tool based on the ISAA, which could be translated into Indian languages and has good psychometric properties [15].

## Materials and methods

Institutional Ethics Committee, ABVIMS, Dr. R.M.L. Hospital approved the study (letter no. 191(10/2017)/IEC/ABVIMS/RMLH)/92 dated 5th March, 2020). The letter says that "the IEC discussed and reviewed application seeking extension of the research project titled "Development and validation of screening version of Indian Scale for assessment of Autism" was discussed in its meeting dated 07/02/2020 and 'extension is granted for a period of one year from 07/02/2020 to 06/02/2021". Written consent forms from the parents of the participating children and written assent form from the children were obtained.

The study was planned in phases [15]. This paper describes the methods and results of the first and second phases which are complete. The study is being conducted at a tertiary care free teaching government institution from April 2019.

### Phase 1

From the original ISAA database, we identified items scored as frequently, mostly or always among participants with autism. We extracted these items and prepared a ten-item screening questionnaire with binary answers, because the binary answer format outperforms the popular seven-point multi-category format with respect to stability, concurrent validity, and speed of completion [34]. The tool needed to be simple enough to be scored by minimally trained workers. We determined face validity with inputs from mental health experts (inclusion criteria below), tested these items on a small sample and finalized the tool.

### Phase 2

The questionnaire formulated in phase 1 was tested among participants registered at our tertiary care free teaching psychiatry outpatient department (OPD), and psychometric properties were determined including reliability, validity and positive predictive value.

### Participant inclusion and exclusion criteria

The experts who participated in the initial evaluation were trained mental health experts including psychiatrists, board certified clinical psychologists, and psychiatric social workers experienced in diagnosing persons with autism. Participants on whom the tool was tested were parents/caregivers of individuals with autism or other psychiatric disorders clinically diagnosed by the treating clinician (gold standard), the autism individual being aged between 3–18 years with no comorbid physical or psychiatric disorder.

### Procedure

Parents/caregivers of all the children reporting to child guidance clinic or general OPD for psychological evaluation were asked to participate in the study. If they agreed, they were referred to the research department where researchers obtained written informed consent

from the parents, and assent from the child if appropriate. Sociodemographic details were obtained; IASQ and thereafter ISAA was administered with due observation and appropriate interview, by two separate trained research workers. Children with profound intellectual disability were excluded. The questions were asked verbatim to the parents and they replied as yes/no. If they did not understand the question additional probes (S1 Appendix) were used. Administration time for IASQ was approximately 10 minutes while the ISAA takes at least one hour.

## Instruments

Socio-demographic, prenatal, perinatal and postnatal information, the tool under evaluation (IASQ) and Indian Scale for Assessment of Autism (ISAA) (from the ISAA manual) were used [33].

## Indian scale for assessment of autism (2009)

Demographic and clinical details were obtained as per the ISAA manual [33]. The ISAA is a 40-item scale divided into six domains- Social Relationship and Reciprocity (9 questions); Emotional Responsiveness (5 questions); Speech-Language and Communication (9 questions); Behavior Patterns (7 questions); Sensory Aspects (6 questions) and Cognitive Component (4 questions). The scores for each item of ISAA range from 1–5, depending on the intensity, frequency and duration of a particular behavior with the following anchors: score 1 = Rarely (frequency up to 20%), score 2 = Sometimes (frequency 21–40%), score 3 = Frequently (frequency 41–60%), score 4 = Mostly (61–80%), and score 5 = Always (81%-100%). Scoring is based on information from parents and observation of the child. Total ISAA scores range from 40–200. The lowest score represents no symptoms or symptoms which were present only rarely, and the maximum score indicates the most severe presentation of autism. The following categorization of scores is recommended: mild: 70–107, moderate: 108–153, severe:>153. ISAA requires 45–60 minutes to administer by a trained and qualified person. Inter-rater reliability (r > 0.83) as well as test-retest reliability after three months (r > 0.89) was satisfactory. A cut off score of 70 showed high and balanced sensitivity and specificity between autism and the group without psychiatric diagnosis, as well as between autism and the mental retardation (MR) group. Receiver Operator Curve (ROC) analysis confirmed the discriminant ability of ISAA (Area under the curve, AUC = 0.931, SE = 0.009 using the cut off score of 70) [12].

## Data analysis

Data obtained was entered into a data base specially prepared for the purpose by the Indian Council of Medical Research (ICMR) [35]. Data was cleaned and outliers checked.

## Test psychometrics

Item analysis was carried out to find out the sensitivity and specificity of items in order to ascertain how well the items differentiated between autism and non-autism. Sensitivity, specificity and area under the ROC curve were calculated using SPSS version 21 [36]. Positive and negative predictive values (PPV, NPV) were calculated by the formulae:

PPV = Number of true positive / (Number of true positives + Number of false positives),

NPV = Number of true negative / (Number of true negative + Number of false negative),

(Positive) Likelihood Ratio LR+ was calculated by using the formula: Sensitivity/ (1-Specificity).

## Reliability, validity

We conducted test–retest and inter-rater reliability.

For test-retest reliability, 40% of participants were proportionately and randomly selected from each score of 1–10 on the IASQ. In case a participant was unreachable or refused to participate, another participant was randomly selected from among those with the same score. The same expert, who had earlier administered the IASQ, re-administered it to the same parent/caregiver within 6 months of the original test. Those who had participated more than six months ago were excluded, to control for real change in symptoms. Intra-class correlation test was used for test-retest reliability.

For inter-rater reliability, 20% of the participants (n = 30) were randomly re-assessed by different raters, blind to each other, on the same day with a gap of at least an hour between the two ratings. Inter rater reliability was calculated using Krippendorff's alpha.

Cronbach's alpha was used to measure internal consistency. Concordance analyses were used for validity.

In addition, descriptive statistics were used to describe sociodemographic variables of the sample.

## Results

The following results are presented in the format proposed by Boateng et al. [37] (Fig 1).

## A. Item development

**A.1 Item development—domain identification and item generation.**   The original dataset of participants with autism from the first ISAA study was used (N = 436, we used 433 whose data was complete). ISAA item scores range from 1–5, based on the duration, intensity and frequency of a particular behavior. The anchor scores are rarely (20%), sometimes (21–40) frequently (41–60%), mostly (61–80%), and always (81%-100%) respectively from score 1 to 5. ISAA has six domains. ISAA items in the autism group answered as frequently present (score 3), mostly present (score 4) and always present (score 5), were identified. A total of 12 most frequently answered positive questions were selected (Table 1) for inclusion and designed to require only a yes or no response.

**A.2. Item development-content validity.**   In a consensus meeting of board-certified psychiatrists, clinical psychologists, and psychiatric social workers experienced in diagnosing autism, experts suggested deletion of ISAA question 38 (shows delay in responding) as this was not specific to autism. ISAA question 6 (unable to respond to social/environmental cues) was not included because the concept was difficult to grasp by laypeople and is very similar to ISAA/IASQ questions 3 (remains aloof) and 4 (does not reach out to others) (Table 1). Two additional questions deemed to be specifically indicative of autism were added–ISAA question 25: shows attachment to inanimate objects; and ISAA question 36: responds to objects unusually by smelling, touching or tasting. Though less than 50% of ISAA database participants responded positively to these questions, our experts felt that that they were typical symptoms of autism and should be included. This was the first draft of the screening tool. The questionnaire was to be answered as yes or no by a parent or caregiver who lived with the child, considering the behavior of the child over time. As the age range in which ISAA had been tested was between 3–18 years, the screening questionnaire was tested in the same age group. The screening instrument was named the Indian Autism Screening Questionnaire (IASQ).

**A.3. Item development—transformation, modification and translation.**   The ISAA includes statements rather than queries. Research experts discussed and reviewed results and suggested that the 10 screening items should be made into questions rather than statements.

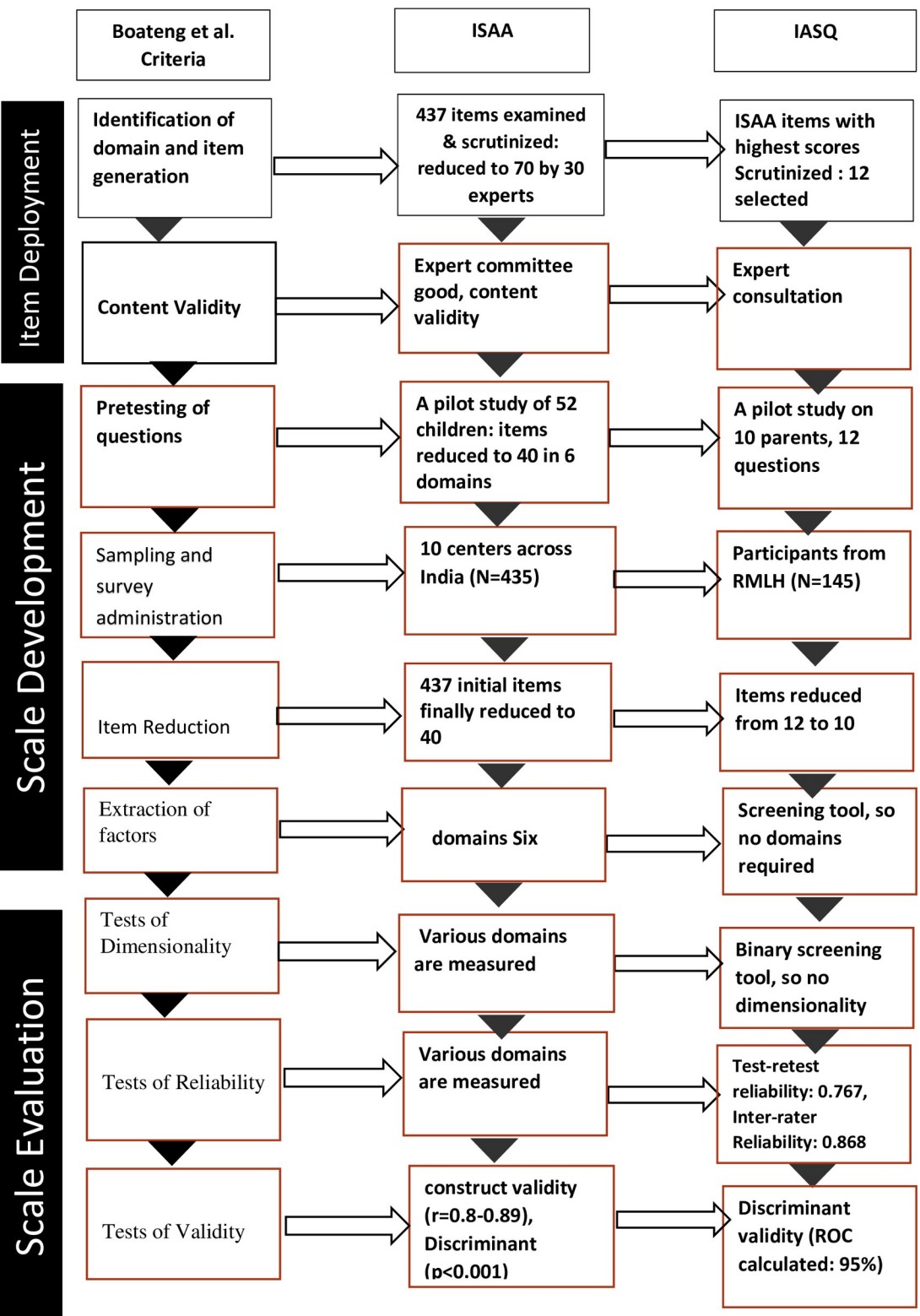

**Fig 1. An overview of the three phases and nine steps of scale development and validation based on Boateng et al (2018).** (Adapted and modified the figure with permission from Dr. G.O. Boateng.).

**Table 1. List of questions and frequency of responses from original ISAA data (N = 433) [*].**

| IASQ Sr. No | ISAA Sr. No | Questions | Frequency % in ISAA dataset |
|---|---|---|---|
| 1 | 1 | Has poor eye contact | 64.90 |
| 2 | 2 | Lacks social smile | 78.06 |
| 3 | 3 | Remains aloof | 75.98 |
| 4 | 4 | Does not reach out to others | 66.97 |
| 5 | 7 | Engages in solitary and repetitive play activities | 73.90 |
| 6 | 9 | Does not maintain peer relationships | 84.53 |
| 7 | 20 | Unable to initiate or sustain conversation with others | 69.52 |
| 8 | 24 | Engages in stereotyped and repetitive motor mechanisms | 69.98 |
| 9 | 25 | Shows attachment to inanimate subjects | 45.50 |
| 10 | 36 | Responds to objects unusually by smelling, touching or tasting | 45.03 |
| Not specific to autism | 38 | Shows delay in responding | 66.05 |
| Similar to questions 3, 4 | 6 | Unable to respond to social/environmental cues | 70.90 |

[*]The last two questions were subsequently not included.

Following this, the 10 items were transformed into questions using the same or similar words; then translated into Hindi and back-translated into English. The questions were further discussed extensively in a second expert meeting with psychiatrists, clinical psychologists and psychiatric social workers, consensually selecting the most appropriate Hindi words for each item. All 10 questions for the screening tool were prepared. The final IASQ with probe questions is attached as S1 Appendix.

## B. Scale development

**B.1. Scale development–pretesting.** Experts were requested to administer the questionnaire to a few parents to check for understanding and acceptability. Research staff also administered the questionnaire on 10 parents of persons with clinically diagnosed autism. Considering their answers, in a subsequent meeting of experts some Hindi words were modified.

*Supplementary explanatory probes.* The National Institute for Mentally Handicapped, Secunderabad (now the National Institute for Empowerment of Persons with Disability-NIEPD) has extensive experience in using the ISAA. They had prepared simple explanations of ISAA items in case the parent did not comprehend the standard questions. These explanations were translated into Hindi. If a parent failed to understand the verbatim item, these supplementary probes could be used by the questioner (S1 Appendix).

**B.2. Scale development—sampling and survey administration: Testing in a clinic population.** According to Boateng et al., potential scale items should be administered on a sample that reflects range of target population They recommended sample sizes of 10 respondents per survey item [37]. Thus, 100 participants were projected; we tested a total of 145 participants (90 with autism and 55 with other psychiatric disorders) (Table 2). There were significantly greater number of participants who were males in both groups but both groups were similar on age and type of schooling. A significantly larger number of participants with autism were currently attending special schools (Chi square = 9.97, p = 0.04). Annual family income of parents of participants with autism was significantly higher than that of the other group (t = 3.68, p = 0.0003).

**B.3. Scale development—item reduction and factor extraction.** As described above, items were reduced from 12 to 10. The ISAA has six domains. The IASQ contains questions

**Table 2. Socio-demographic details and ISAA diagnosis of the sample (N = 145).**

| Group ➡ Variable | Autism on ISAA | Not-Autism on ISAA | Total | Chi square/t-value | p-value |
|---|---|---|---|---|---|
| | N = 90 | N = 55 | N = 145 | | |
| Gender (Male/Female) | 73(81%)/17(19%) | 40 (72.7%)/15 (27.3%) | 113 (77.9%)/32(22.1%) | 1.83 | 0.21 |
| Age in years (Mean ± SD) | 7.95 ± 4.37 | 9.22 ± 4.29 | 8.40 ± 4.36 | -1.55 | 0.12 |
| Currently Studying (Yes/No) | 70 (77.8%)/20(22.2%) | 48(87.3%)/7(12.7%) | 118(81.37%)/27(18.62%) | 2.03 | 0.19 |
| Type of School* 1/2/3/4/ | 48(53.3%)/10(11%)/14 (15.6%)/18(20%) | 43(78.2%)/2(3.6%)/3(5.5%)/ 7(12.7%) | 91(62.8%)/12(8.1%)/25 (17.2%)/17(11.7%) | 9.97 | 0.04 |
| Annual Family Income in Rupees (Mean ± SD) | 818590.91±728784.42 | 402745.45±519675.73 | 658650.35±685074±01 | 3.68 | 0.0003 |
| Identification of autism by IASQ/ISAA/ Clinical diagnosis** | 110/90/91 | 35/55/54 | 145/145/145 | 0.011 | 1.00 |

*General/Integrated and open/Special/Data not available.

** Clinical diagnosis using ICD 10 by clinician.

which broadly cover all domains. Factor extraction was not attempted as this was a screening tool where even one positive answered warranted referral to a higher facility for more detailed testing.

## C) Scale evaluation

**C.1 Scale evaluation—test of dimensionality.** Not attempted as this was a brief screening tool with binary scoring.

**C.2. Scale evaluation—tests of reliability.** *a) Sensitivity & specificity analysis.* The IASQ is a 10-point screening instrument, where the sensitivity and specificity were computed at each cutoff to determine the best threshold (Table 3).

**Sensitivity and Positive Predictive Value (PPV) analysis.** At a cutoff of 1, sensitivity was 99%, specificity 62%, PPV 81%, NPV 95%. At a cutoff of 2, these values were 98%, 71%, 85% and 89% respectively. Calculating positive and negative likelihood ratios, either cutoff (1 or 2) were acceptable, depending on the aim. During a subsequent ICMR Review Meeting (September 2019), whether to adopt 1 or 2 as cutoff was extensively discussed. The overarching aim of this project was to design a screening instrument which would not miss even one child with autism when used in the community. Moreover, even those participants who were 'false positives' were not typical but had received a diagnosis of some other childhood psychiatric disorder which could be diagnosed at the second stage of detailed expert evaluation after screening.

**Table 3. Sensitivity, specificity, positive predictive value and negative predictive value at different cut-off points of IASQ.**

| Cut-off point | Sensitivity | Specificity | Positive predictive value | Negative predictive value | LR+ | LR- |
|---|---|---|---|---|---|---|
| 1 | 0.989 | 0.618 | 0.81 | 0.95 | 2.59 | 0.02 |
| 2 | 0.978 | 0.709 | 0.85 | 0.89 | 3.36 | 0.03 |
| 3 | 0.944 | 0.800 | 0.89 | 0.85 | 4.72 | 0.07 |
| 4 | 0.911 | 0.818 | 0.89 | 0.74 | 5.01 | 0.11 |
| 5 | 0.822 | 0.836 | 0.89 | 0.71 | 5.02 | 0.21 |
| 6 | 0.778 | 0.891 | 0.92 | 0.60 | 7.13 | 0.25 |
| 7 | 0.633 | 0.909 | 0.93 | 0.52 | 6.97 | 0.40 |
| 8 | 0.467 | 0.982 | 1 | 0.38 | 25.67 | 0.54 |
| 9 | 0.144 | 1.000 | 1 | | | 0.86 |
| 10 | 0.044 | 1.000 | 1 | | | 0.96 |

Out of a total of 145, only 4 participants were misdiagnosed as IASQ positive (clinically, one participant was diagnosed as Specific Developmental Disorder, 2 were diagnosed as Intellectual Disability and one was diagnosed as a behavior problem). Thus, for community screening, a cutoff score of 1 was decided by consensus. However, if a research study needs still higher specificity, a higher cut off may need to be used.

**Test-retest reliability.** Using the intra-class correlation test, mean ± standard deviation score for IASQ was 4.45± 3.49 and retest was 4.53±3.46 respectively with r = 0.767 for single measure (CI = 0.62–0.86) (tested for each question) and r = 0.868 for average measure (CI = 0.767–0.925) (mean of all questions). There was no significant difference between baseline IASQ scores and retest IASQ scores on any of the items as well as total scores on paired t-test.

**Inter-rater reliability.** Two experienced raters re-administered the IASQ on 20% of participating parents/caregivers. Krippendorff's alpha was 0.87, indicating strong inter-rater reliability. Krippendorff alpha ignores missing data entirely and can handle various sample sizes. This is a ratio between observed disagreement to expected disagreement.

**Receiver operating characteristics.** We carried out Receiver Operating Characteristic Curve (ROC) analysis to assess the discriminant power of IASQ, using sensitivity and specificity levels at different cut off points (Fig 2). The overall measure of agreement between criterion

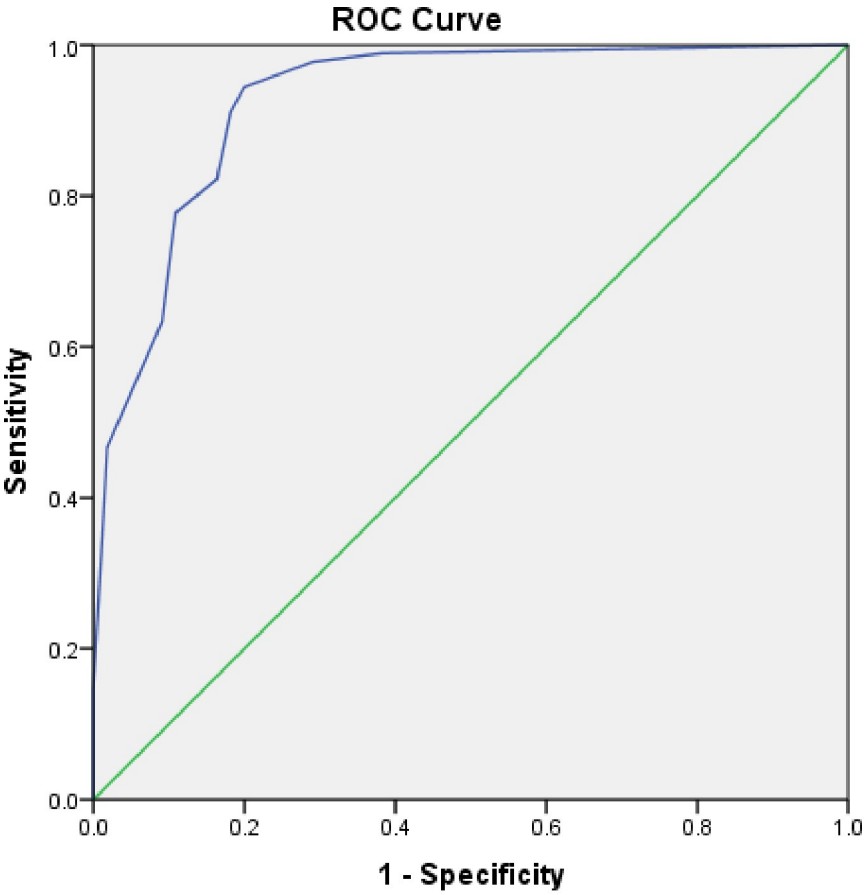

Diagonal segments are produced by ties.

**Fig 2. Results of ROC curve analysis.**

ISAA and IASQ is the area under the curve (AUC). The area under ROC is 95%. We also calculated the sensitivity and specificity for different age groups to check the consistency of the screening instrument (S1 and S2 Figs). The age wise distribution of sensitivity, specificity, true positive, true negative, false positive and false negatives is presented in S1 Table.

For the age groups 3–5, 6–8, 8–11 and 12–14 sensitivity was 100% and specificity was 63%, 100%, 47% and 80% respectively; for 15–18 years where the sample size was lower, the sensitivity was 90% and specificity was only 43%. Out of a total 17 participants in the highest age group, 4 were false positive (clinically, one participant was diagnosed as Specific Developmental Disorder, 2 were diagnosed as Intellectual Disability and one was diagnosed as a behavior problem). A specific and large sample of older participants should be recruited in future studies.

**C.3. Scale evaluation-tests of validity.**    Validation and concordance with clinical diagnoses: Clinical records of the participants, were checked for clinical diagnosis. There were 91 participants with a clinical diagnosis of autism, while 54 participants had a clinical diagnosis other than autism. These diagnoses were compared with IASQ scores above cutoff. Mean IASQ score of participants with autism (6.72 ± 2.12) and non-autism (1.32 ± 2.23) were significantly different (t = 14.57, p<0.0001).

The diagnoses at various cut off points of IASQ were compared with the clinical diagnoses. Their concordance was computed using Cronbach's alpha for all cutoffs which are high till cutoff 8 (S2 Table). It appears that potentially, questions 9 and 10 do not contribute significantly in screening for autism.

## Discussion

Considering the reported increase in the number of persons with autism [38], desirability of early intervention, and obtaining their legal rights through identification and certification it is important to identify all persons autism in the community [39, 40]. Awareness of autism is inadequate in India. Hence there is a need for a simple, brief screening tool which can be used by minimally trained healthcare workers and which can potentially identify all suspected cases. In addition, the screen should possess satisfactory psychometric properties, free for use, should be culturally consonant, easy to translate into Indian languages, and easy to score. Considering all these factors, we opted to extract a questionnaire from the free-for-use ISAA which was developed and is widely used in India.

Using the ISAA database, we identified ten questions which were most commonly positive among cases with autism. To increase its face validity, experts were extensively consulted in finalizing the items- similar questions were deleted while two questions, which were particularly typical of autism, were included. The final IASQ has ten questions with yes/no answers, with a minimum administration time of five minutes. Other screening instruments are longer: the Social Communication Questionnaire -SCQ- has 40 items, Modified M-CHAT for toddlers has 23 items [18, 41, 42] and the original Chandigarh Autism Screening Instrument (CASI) has 37 items [32].

Thabtah and Peebles suggested that while evaluating and designing a screening tool not only are sensitivity, and specificity important, critical issues related to administration, efficiency, target audience, complexity, digital existence, and accessibility should also be considered [43]. Administration of the IASQ is easy, it can be used by non-specialists, it will be free for use after development, and a digital as well as smartphone version is being developed [35]. Thabtah and Peebles also suggest that new tools should correlate with the DSM-5 diagnostic criteria, which are different from DSM-IV criteria [44]. The IASQ items can be correlated with diagnostic criteria of the DSM-5 [45] (Table 4). The IASQ is time efficient as it takes only 5–15

**Table 4. DSM 5 criteria and IASQ items.**

| DSM 5 | IASQ questions |
|---|---|
| Category A: Persistent deficits in social communication and social interaction across multiple contexts, as manifested by the following, currently or by history<br>1. deficits in social communication and social interaction<br>2. deficits in nonverbal communication behaviours<br>3. Deficits in developing, maintaining, and understanding relationships | Qs. 3. Does your child remain aloof?<br>Qs. 7. Is your child unable to initiate or sustain conversation with others?<br>Qs 2. Does your child lack social smile?<br>Qs 1. Does your child have poor eye contact<br>Qs 4. Does your child not reach out to others? |
| Category B: Restricted, repetitive patterns of behaviour, interests, or activities, as manifested by at least two of the following, currently or by history<br>1. Stereotyped or repetitive motor movements, use of objects, or speech<br>2. Insistence on sameness, inflexible adherence to routines, or ritualized patterns<br>3. Highly restricted, fixated interests that are abnormal in intensity or focus.<br>4. Hyper- or hyporeactivity to sensory input or unusual interest in sensory aspects<br>5. Deficits in developing, maintaining, and understanding relationships | Qs 8. Does your child engage in stereotyped and repetitive motor mannerisms?<br>Qs 5. Does your child engage in solitary and repetitive play activities?<br>Qs 9. Does your child show attachment to inanimate object/s?<br>Qs 10. Does your child respond to objects/people unusually by smelling, touching or tasting?<br>Qs 6. Does your child not maintain peer relationships? |

minutes to administer. IASQ is comprehensible as its language is simple, and culturally acceptable (the ISAA was also developed keeping cultural issues in mind).

According to Boateng et al., there are nine steps of scale development [37]. The first is identifying items, domains and defining them. IASQ was extracted from a valid and reliable diagnostic instrument, and its items and domains have been specified, defined, and described in detail in the manual [33]. The second is to evaluate each item. The original instrument has good construct validity (r = 0.8–0.89), and we discussed item selection extensively with mental health experts. In the third step, pre-testing was carried out by clinical psychologists separate from the research staff, and words were simplified after pretesting on a small sample. The next step- test administration on participants with and without the disorder- was carried out with more than adequate sample size (at least 10 participants per item). Step five- item reduction- was not required as the scale was already extracted from a longer version. Boateng et al. suggested factor analysis as the sixth step [37], instead of which we had initially selected questions with the highest positive responses from the ISAA. The IASQ has satisfactory test-retest (0.87) and inter-rater (0.872) reliability as compared to test-retest and inter-rater reliability of ISAA which were 0.93–0.99 and 0.99, respectively. Validation against clinical diagnosis and against the ISAA itself is reasonably high. Validity and correlation analyses with known measures are being carried out.

Parents are reliable sources of information about their children's development and behavior [46]. Evidence-based screening tools that incorporate parent reports can facilitate structured communication between parents and providers to discover parent concerns, increase parent and provider observations of the child's development, and increase parent awareness. Our instrument relies on parents' information and is time and cost efficient.

Future plans include community survey, and induction and training of primary care workers in order to test the instrument in the community.

## Strengths

Tools that are developed in high income countries may be problematic in terms of cultural norms, copyright permission and payment for translation into other languages [47]. Tools

used to screen for autism in low-and-middle income countries are often derived from these same existing tools [17]. The ISAA was developed keeping these issues in mind, as also the IASQ. In addition, there is potential room for improvement, for example by creating variants that are better suited for the older children and as yet unidentified adults. The IASQ is quite comprehensive for younger children and it is important to screen and identify at younger age for timely intervention.

## Limitations

Validation was done against the ISAA a tool from which the IASQ itself is derived. As a gold standard, we validated the tool against clinical diagnosis by experts but are now testing it against the Childhood Autism Rating Scale (CARS) [19]. It will also be tested in the community and in other areas where persons with autism (a comparatively rare population diagnosis) will be easier to recruit. There was significant difference between autism and non-autism children on family income suggesting autism children reported from higher socioeconomic strata. Although we had recruited adequate sample (at least 10 participants per item) there were relatively fewer older individuals. In subsequent studies a larger sample for this group should be included to make this instrument more applicable to people of all ages.

## Conclusions

The ISAA is a reliable and validated instrument that can be used for community-based screening campaigns for Autism in Hindi speaking regions of India. Its utility in other settings should be evaluated.

## Supporting information

**S1 Fig. An overview of the three phases and nine steps of scale development and validation.**
(PDF)

**S2 Fig. Age-wise sensitivity and specificity of IASQ.**
(DOCX)

**S1 Table. Age-wise distribution of true positive, true negative, false positive and false negative cases with sensitivity and specificity at cut off score of 1.**
(DOCX)

**S2 Table. Concordance of IASQ diagnoses at various cutoff points with clinical diagnoses of autism and non-autism participants.**
(DOCX)

**S1 File.**
(XLSX)

**S2 File.**
(PDF)

**S1 Appendix.**
(DOCX)

## Acknowledgments

We thank Dr. Soumya Swaminathan (then Secretary, Dept. of Health Research, DHR), Dr. Balram Bhargav (Secretary DHR), and Dr. Harpreet Singh, ICMR. We thank the faculty of

'Cross-Fertilized Research Training for New Investigators in India and Egypt' (D43 TW009114). We thank the National Coordinating Unit for logistic support and the ICMR Data Management Unit for designing the database.

## Author Contributions

**Conceptualization:** Satabdi Chakraborty, Triptish Bhatia, Jaspreet S. Brar, Vishwajit L. Nimgaonkar, Smita Neelkanth Deshpande.

**Data curation:** Vikas Sharma, Nitin Antony, Dhritishree Das, Sushree Sahu, Satyam Sharma, Vandana Shriharsh.

**Formal analysis:** Triptish Bhatia, Vikas Sharma, Satish Iyengar.

**Investigation:** Nitin Antony, Dhritishree Das, Sushree Sahu, Satyam Sharma, Vandana Shriharsh.

**Methodology:** Triptish Bhatia, Satyam Sharma, Smita Neelkanth Deshpande.

**Project administration:** Satyam Sharma, Vandana Shriharsh, Smita Neelkanth Deshpande.

**Resources:** Satabdi Chakraborty, Ravinder Singh.

**Software:** Jaspreet S. Brar, Satish Iyengar.

**Supervision:** Satabdi Chakraborty, Satyam Sharma, Vandana Shriharsh, Smita Neelkanth Deshpande.

**Validation:** Vikas Sharma, Sushree Sahu, Satish Iyengar, Vishwajit L. Nimgaonkar.

**Visualization:** Dhritishree Das, Sushree Sahu, Ravinder Singh.

**Writing – original draft:** Triptish Bhatia, Nitin Antony.

**Writing – review & editing:** Satabdi Chakraborty, Jaspreet S. Brar, Satish Iyengar, Ravinder Singh, Vishwajit L. Nimgaonkar, Smita Neelkanth Deshpande.

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
