## [Decision Letter · Decision Letter 0]

4 Mar 2021

PONE-D-20-39733

Psychometric properties of a screening tool for autism in the community- the Indian Autism Screening Questionnaire (IASQ)

PLOS ONE

Dear Dr. Deshpande,

Thank you for submitting your manuscript to PLOS ONE. We believe that the paper may be suitable for publication in Plos One, but would benefit from a revision. Therefore, we invite you to submit a revised version of the manuscript that addresses the points raised during the review process.

Please below find the comments of the two reviewers. Please check your manuscript for language errors and typos. Also please delete the subheadings (underlined) in the Introduction section. You may also add a research question at the end of the Introduction. Be consistent with regard to headings. You use headings in the Discussion section, but you do not use headings (Data analysis, Procedure, Participants etc) in the Method section. Thus, take a look at the structure of the paper and be consistent between major sections.

We look forward to receiving your revised manuscript.

Kind regards,

Robert Didden

Academic Editor

PLOS ONE

Journal Requirements:

2. Please include a copy of the validate questionnaire in the original language, as Supporting Information, or include a citation if it has been published previously.

3. Please discuss how the original Indian Scale for Assessment of Autism was validated prior to use in India.

5. Please amend the manuscript submission data (via Edit Submission) to include author Ravinder Singh.

6. Please include a separate caption for each figure in your manuscript.

7. Please include captions for your Supporting Information files at the end of your manuscript, and update any in-text citations to match accordingly. Please see our Supporting Information guidelines for more information: http://journals.plos.org/plosone/s/supporting-information

Reviewers' comments:

Reviewer's Responses to Questions

**Comments to the Author**

1. Is the manuscript technically sound, and do the data support the conclusions?

Reviewer #1: Yes

Reviewer #2: Yes

2. Has the statistical analysis been performed appropriately and rigorously? 

Reviewer #1: Yes

Reviewer #2: Yes

3. Have the authors made all data underlying the findings in their manuscript fully available?

Reviewer #1: No

Reviewer #2: Yes

4. Is the manuscript presented in an intelligible fashion and written in standard English?

Reviewer #1: Yes

Reviewer #2: No

5. Review Comments to the Author

Reviewer #1: Dear authors,

Thank you for the possibility to read your manuscript "Psychometric properties of a screening tool for autism in the community- the Indian Autism Screening Questionnaire (IASQ)". It describes an important area in autism spectrum disorders (ASD), and screening as an important factor in other countries and cultures.

The manuscript is describing the study in a good manner, however I have some comments;

- The manuscript states that "All children should be screened in the community at an early age". There are some scientific discussion on the efficacy of universal screening, and it is unclear what you base this statement on?

- The decision to use a binary scoring for each item is not described in the manuscript. As autism may be shown as qualitative differences in functioning, other studies has used a vider scale, and I wonder why this study did not?

- The use of a cut-off score of 1 (out of 10), man pose risk for a high portion of false positives - even if this did not in this selected sample. This risk and possible consequences and remidation may be discussed to give the reader a better understanding

Reviewer #2: There were few grammatical errors in the abstract and in the introduction (ex. line 117 add "is"; line 118 add "a"). Perhaps also use "individuals" instead of "persons", or older children?

one more thing to consider: the article is very nicely done. However the populaiton studied is relatively small. Perhaps mentioning this as a limitation as well, so to generalize in larger groups.

6. PLOS authors have the option to publish the peer review history of their article (what does this mean?). If published, this will include your full peer review and any attached files.

Reviewer #1: **Yes: **Kenneth Larsen

Reviewer #2: No

---

## [Author Response · Author response to Decision Letter 0]

18 Mar 2021

Dr. Smita N. Deshpande, MD, DPM

Professor,

Dept. of Psychiatry, 

Centre of Excellence in Mental Health,

Atal Bihari Vajpayee Institute of Medical Sciences 

& Dr Ram Manohar Lohia Hospital,

Banga Bandhu Sheikh Mujib Road,

New Delhi 110001

March 11, 2021

To,

Dr. Emily Chenette

Editor –in-Chief

PLOS ONE.

Attn.: Dr. Robert Didden

Ref.: Revised submission of our manuscript number PONE-D-20-39733: “ Psychometric properties of a screening tool for autism in the community- the Indian Autism Screening Questionnaire (IASQ)”

Dear Editors,

Thank you for getting our manuscript reviewed. The reviewers’ comments are thoughtful and very encouraging, as they understood the need for an instrument for community screening for autism in LMIC.

We are resubmitting our modified manuscript as per the Academic Editor’s and Respected Reviewers’ comments. 

We have no legal or ethical issues for uploading anonymized data of our study, hence we have uploaded our study data along with the revised manuscript. We have deleted the subheadings in the introduction section and added the research question at the end of the introduction and methods. We have uploaded the response to reviewers, marked and unmarked copies of the modified manuscript. There is no change in financial disclosure. Figure files are also uploaded according to PLOS One format. 

We hope this modified manuscript is according to the journal’s expectations and will be accepted for publication.

Thank you for an early response and acceptance if possible!

Sincerely,

Smita N Deshpande

Reply to comments

PONE-D-20-39733: “ Psychometric properties of a screening tool for autism in the community- the Indian Autism Screening Questionnaire (IASQ)”

Journal Requirements:

 Reply1: The revised manuscript follows all PLOS One style requirements.

Query 2: Please include a copy of the validated questionnaire in the original language, as Supporting Information, or include a citation if it has been published previously.

Reply: Thank you for this suggestion. We have included the bilingual IASQ questionnaire as appendix.

Query 3. Please discuss how the original Indian Scale for Assessment of Autism was validated prior to use in India.

Reply 3: We have added the following text in the introduction section (see below for the entire paragraph on the ISAA):

The Indian Scale for Assessment of Autism (ISAA) was developed by the Government of India for detailed assessment of autism and is mandated for evaluating degree of disability. It is a 40-item tool which can be used for certification and follow-up but not for screening [30]. ISAA was developed and validated in a multi-centric Pan-Indian study comprising 1124 participants aged 3-22 years with clinically diagnosed Autism/Pervasive Developmental Disorder (PDD-ICD10), Intellectual Disability (ID), other psychiatric disorders and persons with normal intellect. The scale was compared with Childhood Autism Rating Scale (CARS) (Version 1) [31]. Reliability and internal consistency of ISAA as compared to CARS was satisfactory [30, 31]. Validity of ISAA was determined through content validity and criterion validity. Initially an item pool of 437 items was generated, shortened to 57 after expert inputs from 30 experts, and then to 40 after a pilot study. At the end of the main study, each individual item was correlated with total ISAA scores which was significant at 0.001 level suggesting that all items were valid in differentiating between autism and other groups. The autism group had significantly higher scores than other groups suggesting high discriminant value. The criterion test validity of ISAA was determined by comparing total scores on ISAA with those on CARS using Pearson Product Moment Correlation (r=0.765, p<0.001). Internal consistency and reliability (Cronbach's coefficient alpha) were significant and comparable to CARS (Cronbach's alpha 0.932 P < 0.001) [31].However, the ISAA is a detailed evaluation tool which takes time and training, and cannot be used for population studies.

Reply 4: There are no ethical or legal restrictions. We have uploaded the data now.

 Query 5. Please amend the manuscript submission data (via Edit Submission) to include author Ravinder Singh.

Reply 5: We apologize for this omission. We have added his name now.

 Query 6. Please include a separate caption for each figure in your manuscript.

Reply 6: We have added the separate captions.

Query 7. Please include captions for your Supporting Information files at the end of your manuscript, and update any in-text citations to match accordingly. Please see our Supporting Information guidelines for more information: http://journals.plos.org/plosone/s/supporting-information

Reply 7: Captions for supporting information files are included.

Reviewers' comments:

Reviewer's Responses to Questions

Comments to the Author

1. Is the manuscript technically sound, and do the data support the conclusions?

Reviewer #1: Yes

Reviewer #2: Yes

2. Has the statistical analysis been performed appropriately and rigorously?

Reviewer #1: Yes

Reviewer #2: Yes

3. Have the authors made all data underlying the findings in their manuscript fully available?

Reviewer #1: No

Reviewer #2: Yes

Note: We have now uploaded all our study data.

4. Is the manuscript presented in an intelligible fashion and written in standard English?

Reviewer #1: Yes

Reviewer #2: No

 Note: Thanks to our expert reviewers for their above remarks.

5. Review Comments to the Author

Reviewer #1: Dear authors,

Thank you for the possibility to read your manuscript "Psychometric properties of a screening tool for autism in the community- the Indian Autism Screening Questionnaire (IASQ)". It describes an important area in autism spectrum disorders (ASD), and screening as an important factor in other countries and cultures.

Reply: Thank you for your encouraging words. This is indeed an important first step regarding intervention planning for children with autism.

The manuscript is describing the study in a good manner, however I have some comments;

Query 1- The manuscript states that "All children should be screened in the community at an early age". There are some scientific discussion on the efficacy of universal screening, and it is unclear what you base this statement on?

 Reply 1: We agree entirely with the expert reviewer, that universal early screening may identify false positives. However, it has significant advantages where healthcare services are less easily available. We have added the following text in introduction: 

Rashtriya Bal Swasthya Karyakram (RBSK) is a flagship Government of India Programme focused on early identification and intervention for children from birth to 18 years. It includes early identification of 4 ‘D’s viz. Defects at birth, Deficiencies, Diseases, Development delays including disability (including autism). [11]. For early identification all children should be screened in the community at a younger age. Community screening has several advantages [12].

Query 2: - The decision to use a binary scoring for each item is not described in the manuscript. As autism may be shown as qualitative differences in functioning, other studies has used a wider scale, and I wonder why this study did not?

 Reply 2: The study is aimed at screening the population by community health workers, who should be trained minimally. The screening instrument should be simple and easy to use by non-specialists. The diagnostic instrument ISAA uses wider scale as it should be used by expert trained workers. We have added the text in introduction for binary scoring as follows:

We extracted these items and prepared a ten-item screening questionnaire with binary answers, because the binary answer format outperforms the popular seven-point multi-category format with respect to stability, concurrent validity, and speed of completion [34]. The tool needed to be simple enough to be scored by minimally trained workers.

Query 3: - The use of a cut-off score of 1 (out of 10), may pose risk for a high portion of false positives - even if this did not in this selected sample. This risk and possible consequences and remediation may be discussed to give the reader a better understanding

 Reply 3: We agree with our respected expert reviewer that cut off score of 1 may increase false positives but after screening, children need to be referred to a higher diagnostic centre and final diagnosis be made after expert review. In future, depending on the requirements, authors could use a different cut off. We have added a line in our revision thus:

The overarching aim of this project was to design a screening instrument which would not miss even one child with autism when used in the community. Moreover, even those participants who were ‘false positives’ were not typical but had received a diagnosis of some other childhood psychiatric disorder which could be diagnosed at the second stage of detailed expert evaluation after screening. Out of a total of 145, only 4 participants were misdiagnosed as IASQ positive (clinically, one participant was diagnosed as Specific Developmental Disorder, 2 were diagnosed as Intellectual Disability and one was diagnosed as a behavior problem). Thus, for community screening, a cutoff score of 1 was decided by consensus. However, if a research study needs still higher specificity, a higher cut off may need to be used. 

Reviewer #2: There were few grammatical errors in the abstract and in the introduction (ex. line 117 add "is"; line 118 add "a"). Perhaps also use "individuals" instead of "persons", or older children?

Reply1: Thank you for pointing this out. We have changed the grammar accordingly.

Query 2: one more thing to consider: the article is very nicely done. However, the population studied is relatively small. Perhaps mentioning this as a limitation as well, so to generalize in larger groups.

Reply2: Thank you for appreciation. We have added the sample size issue as a limitation. Though our sample size was adequate for younger children, we needed a larger sample for older individuals. We have added the following text in limitations:

Although we had recruited adequate sample (at least 10 participants per item) there were relatively fewer older individuals. In subsequent studies a larger sample for this group should be included to make this instrument more applicable to people of all ages.

---

## [Decision Letter · Decision Letter 1]

29 Mar 2021

Psychometric properties of a screening tool for autism in the community- the Indian Autism Screening Questionnaire (IASQ)

PONE-D-20-39733R1

Dear Dr. Deshpande,

We’re pleased to inform you that your manuscript has been judged scientifically suitable for publication and will be formally accepted for publication once it meets all outstanding technical requirements.

Kind regards,

Robert Didden

Academic Editor

PLOS ONE

Additional Editor Comments (optional):

Reviewers' comments:

Reviewer's Responses to Questions

**Comments to the Author**

1. If the authors have adequately addressed your comments raised in a previous round of review and you feel that this manuscript is now acceptable for publication, you may indicate that here to bypass the “Comments to the Author” section, enter your conflict of interest statement in the “Confidential to Editor” section, and submit your "Accept" recommendation.

Reviewer #1: All comments have been addressed

2. Is the manuscript technically sound, and do the data support the conclusions?

Reviewer #1: Yes

3. Has the statistical analysis been performed appropriately and rigorously? 

Reviewer #1: Yes

4. Have the authors made all data underlying the findings in their manuscript fully available?

Reviewer #1: Yes

5. Is the manuscript presented in an intelligible fashion and written in standard English?

Reviewer #1: Yes

6. Review Comments to the Author

Reviewer #1: Thank you for your comments on the review issues.

I think it is a good manuscript, that contribute to increasing the knowledge of early identification of ASD

7. PLOS authors have the option to publish the peer review history of their article (what does this mean?). If published, this will include your full peer review and any attached files.

Reviewer #1: **Yes: **Kenneth Larsen

---

## [Editor Report · Acceptance letter]

12 Apr 2021

PONE-D-20-39733R1 

Psychometric properties of a screening tool for autism in the community- the Indian Autism Screening Questionnaire (IASQ) 

Dear Dr. Deshpande:

I'm pleased to inform you that your manuscript has been deemed suitable for publication in PLOS ONE. Congratulations! Your manuscript is now with our production department. 

Kind regards, 

on behalf of

Professor Robert Didden 

Academic Editor

PLOS ONE